🔓 | **Open Peer Review** | Bacteriology | Research Article

# Refined methodology for quantifying *Pseudomonas aeruginosa* virulence using *Galleria mellonella*

Christopher M. R. Axline,[1,2] Travis J. Kochan,[1] Sophie Nozick,[1] Timothy Ward,[1,3] Tania Afzal,[1,4] Issay Niki,[1,2] Sumitra D. Mitra,[1] Ethan VanGosen,[1,5] Julia Nelson,[1,2] Aliki Valdes,[1] David Hynes,[1,6] William Cheng,[1] Joanne Lee,[1] Prarthana Prashanth,[1,7] Timothy L. Turner,[1] Nathan B. Pincus,[1] Marc H. Scheetz,[8,9] Kelly E. R. Bachta,[1,10] Alan R. Hauser[1,10]

**ABSTRACT** Larvae of *Galleria mellonella* (the greater wax moth) are being increasingly used as a model to study microbial pathogenesis. In this model, bacterial virulence is typically measured by determining the 50% lethal dose ($LD_{50}$) of a bacterial strain or mutant. The use of *G. mellonella* to study *Pseudomonas aeruginosa* pathogenesis, however, is challenging because of the extreme sensitivity of larvae to this bacterium. For some *P. aeruginosa* strains, as few as 1–5 colony-forming units are sufficient to kill *G. mellonella,* which poses challenges for determining $LD_{50}$ values. For this reason, some groups have used time-to-death as a measure of *P. aeruginosa* virulence, but methodologies have not been standardized. We provide a detailed protocol for using the time at which 50% of larvae have died ($LT_{50}$) at a particular inoculum as a measure of *P. aeruginosa* virulence. We also describe a quality control metric for enhancing the reproducibility of $LT_{50}$ values. This approach provides an accurate and reproducible methodology for using *G. mellonella* larvae to measure and compare the virulence of *P. aeruginosa* strains.

**IMPORTANCE** *Pseudomonas aeruginosa* is a significant cause of morbidity and mortality. The invertebrate *Galleria mellonella* is used as a model to determine the virulence of *P. aeruginosa* strains. We provide a protocol and analytical approach for using a time-to-death metric to accurately quantify the virulence of *P. aeruginosa* strains in *G. mellonella* larvae. This methodology, which has several advantages over 50% lethal dose approaches, is a useful resource for the study of *P. aeruginosa* pathogenicity.

**KEYWORDS** *Pseudomonas aeruginosa*, *Galleria mellonella*, virulence

*G*alleria mellonella, also known as the greater wax moth, is found throughout the world. In nature, this organism has a parasitic relationship with honeybees, infiltrating hives and damaging colonies (1). Commercially, the larvae are reared as food for reptiles and as live bait for fishermen. In recent years, *G. mellonella* has been used extensively as a model system for the study of microbial pathogens (2). The larvae have several advantages over mice as a model system for the study of microbial pathogenesis (2). They are invertebrates and thus facilitate the goal of reducing the use of mammals in research (3). This leads to minimal regulatory burdens and allows for the use of death as an end point. They are inexpensive, easy to manipulate and maintain, and survive at 37°C, which approximates the internal human body temperature, an advantage when studying temperature-regulated virulence factors (4). Some components of the larval immune system, such as hemocytes and antimicrobial peptides, mimic those of the human innate immune system (5). Together, these features make *G. mellonella* an attractive model organism for studying the pathogenesis of certain bacteria.

**Peer Reviewer** Alyssa Walker, University of Florida, Gainesville, Florida, USA

Address correspondence to Alan R. Hauser, ahauser@northwestern.edu.

The authors declare no conflict of interest.

*Pseudomonas aeruginosa* is a gram-negative, opportunistic pathogen that is a prominent cause of infections in hospitalized patients, those with underlying comorbidities, and individuals with cystic fibrosis (6). To facilitate infection, this pathogen uses a myriad of virulence determinants, including a type III secretion system, three type VI secretion systems, type IV pili, quorum sensing, and pyocyanin biosynthesis (6). Population genomics and systems biology approaches are emerging as powerful tools for exploring *P. aeruginosa* pathogenesis and treatment (7). However, these studies often require characterization of the virulence of a large number of *P. aeruginosa* strains in the context of a host, a need for which mice are less well suited than *G. mellonella*.

Most studies that use *G. mellonella* as a model for *P. aeruginosa* infection have measured 50% lethal dose ($LD_{50}$) values to quantify virulence (8–12). However, unlike many other bacterial pathogens, *P. aeruginosa* is extremely pathogenic to *G. mellonella*, making these virulence studies challenging. For example, whereas approximately $10^5$ colony-forming units (CFUs) of *Klebsiella pneumoniae* (13) and $10^6$ CFUs of *Staphylococcus aureus* (14) are required to kill *G. mellonella*, as few as 1–5 CFUs of many strains of *P. aeruginosa* are sufficient for this purpose (12, 15). For this reason, accurate $LD_{50}$ values are sometimes difficult to obtain. The second problem is that reproducible inoculation of such low numbers of bacteria is technically challenging and is frequently associated with large relative errors that may dramatically change reported survival values. Even when inoculums are accurately measured by plating following the experiment, it is difficult to reproducibly prepare the same inoculum size for distinct *P. aeruginosa* strains at the time of injection, making strain-to-strain comparisons of virulence problematic. Recently, some investigators have attempted to circumvent these difficulties by using time-to-death rather than lethal dose as a measure of *P. aeruginosa* virulence in *G. mellonella* (16). In this approach, the 50% lethal time ($LT_{50}$)—the time following inoculation at which 50% of larvae have succumbed from a specified dose of a microbe—is used to quantify virulence. However, standardization and general applicability of this methodology for comparing the virulence of *P. aeruginosa* strains have not been systematically investigated. Here, we provide a protocol for obtaining $LT_{50}$ values in *G. mellonella* to quantify *P. aeruginosa* virulence.

## MATERIALS AND METHODS

### Bacterial strains

*P. aeruginosa* bloodstream isolates PABL038, PABL051, PABL066, PABL083, PABL089, and PABL093 were originally obtained from the Clinical Microbiology Laboratory of Northwestern Memorial Hospital (Chicago, IL, USA) and stored at −80°C. These isolates and their corresponding $LD_{50}$ values in mice were previously published (17, 18).

### The *G. mellonella* model

A detailed experimental protocol describing the *G. mellonella* methodology is provided in the supplemental Methods. Briefly, isolates were grown overnight at 37°C in lysogeny broth (LB) with shaking (250 RPM) and then sub-cultured in LB, pelleted, and resuspended in phosphate-buffered saline (PBS). Estimated doses for injection into larvae were obtained by diluting bacteria in PBS to an optical density at 600 nm ($OD_{600}$) of ~0.2 (the equivalent of ~$5 \times 10^7$ CFU/mL) and making appropriate dilutions with PBS to generate a range of bacterial doses (i.e., a range of suspensions of different CFU/mL) for injection into *G. mellonella*. Unless otherwise noted, larvae (Speedy Worm, Alexandria, MN, USA) were injected at the site of the final proleg with 10 µL of *P. aeruginosa* bacteria suspended in PBS using a Legato 100 Syringe Pump (KD Scientific, Holliston, MA, USA). Each inoculum for each *P. aeruginosa* isolate was injected into 10 larvae (i.e., 10 technical replicates). Injection with PBS alone was used as a negative control. Injected larvae were kept at 37°C in Petri dishes and monitored hourly for death, which was inferred by the absence of movement following gentle shaking. The number of dead larvae was recorded at each time point.

## Analysis and statistics

For each dose and *P. aeruginosa* strain, an $LT_{50}$ value was calculated using the time of death of each of the 10 larvae by applying a custom R-Script utilizing the DRC package (https://github.com/ChrisAxline/Galleria_Code.git) (19). Because *P. aeruginosa* bacteria have been reported to multiply exponentially within *G. mellonella* (12, 15), we anticipated a logarithmic relationship between bacterial dose and time-to-death, so the natural logs of the *P. aeruginosa* doses were plotted against these $LT_{50}$ values. Linear regressions were performed using GraphPad PRISM version 10.2.3 (GraphPad Software, Boston, MA, USA, www.graphpad.com) to fit equations to these data points. Statistical differences between the $LT_{50}$ vs ln(dose) equations of different bacterial strains were determined using PRISM to generate non-overlapping 95% confidence intervals (CIs) of $LT_{50}$ values at doses of 2,000 or 60,000 CFU.

## RESULTS

### Testing multiple doses of *P. aeruginosa* has several advantages in estimating the time to death of *G. mellonella* following infection with *P. aeruginosa*

As a first attempt at adopting a time-to-death strategy to measure *P. aeruginosa* virulence in *G. mellonella*, we injected a single bacterial dose into larvae and monitored mortality over time. We used $OD_{600}$ measurements to prepare approximate *P. aeruginosa* inoculum sizes for injection and then performed plating and colony enumeration of the inoculum to more definitively determine these inoculum sizes—a quantification method that did not yield results until the day after the *G. mellonella* larvae were injected. We encountered two difficulties in using this approach. First, each dose measurement was associated with substantial error, and inoculums of small sizes were prone to especially large relative measurement errors (Fig. S1). These errors had the potential to translate into substantial differences in the times that larvae died, causing problems with reproducibility. Second, optical density vs CFU correlations varied significantly from strain to strain of *P. aeruginosa*, making it difficult to accurately prepare inoculums of the same targeted sizes to compare the lethality of distinct *P. aeruginosa* strains (Fig. S2).

To address these issues, we chose to use a range of doses for each tested strain rather than a single targeted dose, thus minimizing the effect of error associated with any one dose. We then plotted the $LT_{50}$ values vs the natural log of the doses for each strain. An advantage of this approach is that the use of multiple data points allows more accurate estimations of $LT_{50}$ values for very small doses of bacteria. A second advantage is that a regression line fit to multiple data points allows for the estimation of an $LT_{50}$ value for any dose and therefore precludes the need to accurately prepare a dose of a targeted size. For example, one may wish to know the $LT_{50}$ value corresponding to an inoculum of 2,000 CFU. As described above, it can be quite difficult to prepare a bacterial suspension of exactly 2,000 CFU (Fig. S2). However, the regression line fit to the $LT_{50}$ vs ln(dose) data points generates an equation that provides $LT_{50}$ values as a function of ln(dose). The $LT_{50}$ value corresponding to a dose of 2,000 CFU can be estimated from this equation even if the operator was unable to prepare an injection dose of exactly 2,000 CFU.

To illustrate this time-to-death methodology, we applied it to the *P. aeruginosa* bloodstream isolate PABL089. Ten larvae were infected with a dose (12,300 CFU) of PABL089, and larval death was monitored each hour. An $LT_{50}$ value (~14 hours) was then estimated based on the time of death of the 10 larvae (Fig. 1A). This process was repeated with additional doses of PABL089 (485, 705, 790, 1,265, 1,650, 2,867, 3,250, 4,350, and 5,950 CFU). The resulting $LT_{50}$ values were plotted against the natural log of the CFU of each dose (Fig. 1B). The $LT_{50}$ value of PABL089 at an arbitrarily chosen dose could then be calculated from these data (e.g., 18.2 hours for a dose of 2,000 CFU, as shown in Fig. 1B). This example illustrates how the time-to-death methodology can be used to quantify the virulence of *P. aeruginosa* strains.

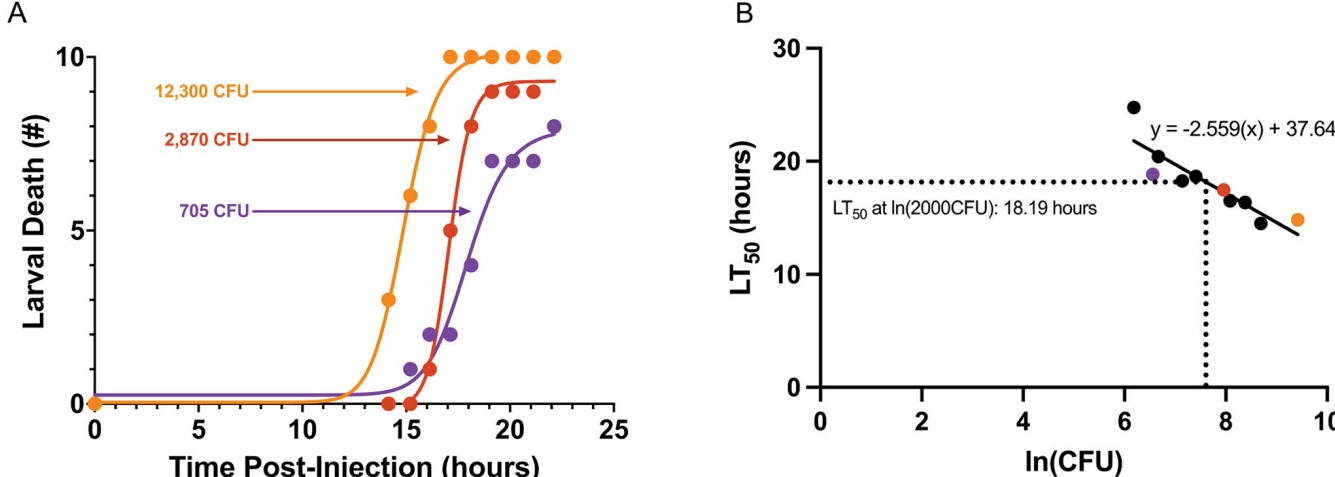

**FIG 1** Estimating the virulence of a *P. aeruginosa* strain using the time-to-death approach. *P. aeruginosa* strain PABL089 was used as an example to demonstrate the LT$_{50}$ approach. (A) Ten *G. mellonella* larvae were injected with each of the *P. aeruginosa* inoculums of 12,300, 2,870, or 705 CFU. Cumulative larval mortality was monitored over time. LT$_{50}$ values were calculated by fitting a sigmoidal curve to mortality observations. (B) LT$_{50}$ values were next measured for additional doses of PABL089, and all LT$_{50}$ data points were plotted against the natural log of their respective doses. Colored dots represent the corresponding doses shown in panel A. A linear regression line was fit to the data points. The virulence of PABL089 at any specific dose was then estimated by interpolating the LT$_{50}$ value corresponding to that dose (18.19 hours at 2,000 CFU in this example).

## The weights of *G. mellonella* larvae affect time to death following inoculation with *P. aeruginosa*

To further enhance reproducibility, we studied the impact of larval weight on LT$_{50}$ values. Prior reports have used larvae of various weights for infections, both in the context of *P. aeruginosa* (20) and other pathogens (21). However, to our knowledge, the impact of weight on virulence readouts has not been systematically investigated. Larvae weighing less than 150 mg were difficult to inject due to their small size and were not tested. Relatively few larvae weighed >300 mg, so those between 300 and 350 mg were combined with those weighing 250–299 mg, and those extremely rare larvae >350 mg were excluded. We injected a series of doses of PABL089 into larvae of the following weight ranges: 150–199, 200–249, and 250–350 mg. LT$_{50}$ values were plotted against the natural log of dose for each weight range (Fig. 2). The differences between the groups were not statistically significant, although the smaller larvae showed a trend toward dying slightly faster. We chose to exclude larvae weighing less than 200 mg and used larvae weighing between 200 and 350 mg for subsequent experiments.

## Small variations in the site of injection do not affect time to death of *G. mellonella* following inoculation with *P. aeruginosa*

We next examined whether slight variations in the site of injection (e.g., due to variability in the technique used by different operators) altered time-to-death measurements, perhaps by causing traumatic injury to critical *G. mellonella* structures. In previous studies, *G. mellonella* were most commonly injected at the final proleg (see the supplemental Methods Protocol, Fig. 4) (21, 22). We examined whether injections with *P. aeruginosa* strain PABL089 slightly (approximately 1 mm) cephalic or caudal to the final proleg or at the site of the proleg affected LT$_{50}$ values. The LT$_{50}$ vs ln(dose) values were similar for the different injection sites, indicating that slight variations in injection sites did not appreciably affect virulence measurements (Fig. 3). For the purpose of consistency, all injections in subsequent experiments were targeted directly to the final proleg.

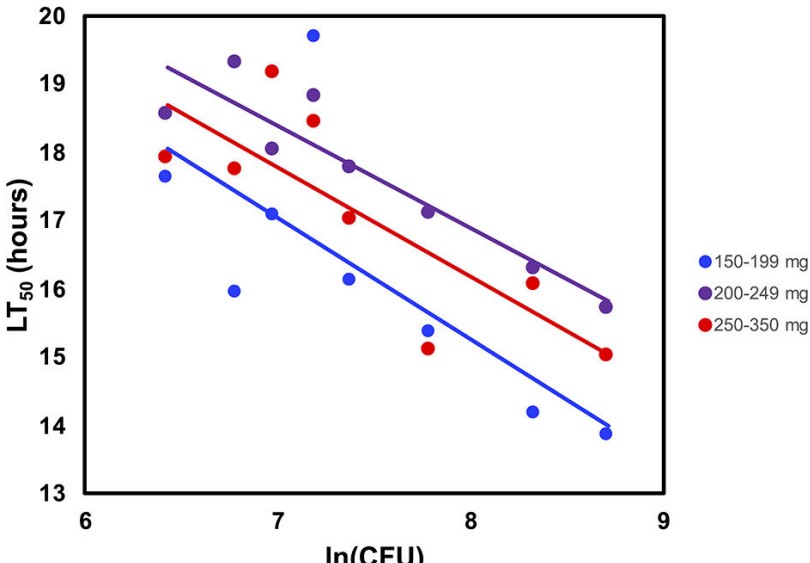

**FIG 2** Effect of *G. mellonella* larval weight on $LT_{50}$ values following injection with *P. aeruginosa*. Larvae weighing 150–199, 200–249, and 250–350 mg were infected with strain PABL089, and $LT_{50}$ values were measured. Each symbol represents the $LT_{50}$ value calculated from 10 larvae given the same dose of PABL089. The experiment was performed twice, and the results were combined. Regression lines were then fit to the data. For the 150–199, 200–249, and 250–350 mg groups, 95% CIs for $LT_{50}$(2,000 CFU) were 14.73–17.20, 15.96–17.68, and 17.04–17.93 hours, respectively; 95% CIs for $LT_{50}$(60,000 CFU) were 3.89–15.94, 7.21–15.58, and 10.26–14.57 hours, respectively.

## Testing of multiple doses allows for the assessment of the quality of time-to-death measurements

Another advantage of testing multiple doses rather than a single dose for each strain is that the results from multiple doses can be used as a quality control metric. Despite standardization of the injection process, as described above, we noted that unexpectedly short or long time-to-death measurements of *G. mellonella* larvae sometimes still occurred, especially when inexperienced operators performed the experiments. An example of such a set of data that was generated with the bloodstream isolate PABL051 is shown in Fig. 4A. We took advantage of the theoretical linear relationship between $LT_{50}$ and ln(dose) to generate a quality control metric that would flag such data sets. For each data set, the scatter in $LT_{50}$ vs ln(dose) was quantified by calculating an $R^2$ coefficient for the goodness-of-fit of an optimal regression line. Those data sets with $R^2$ coefficients < 0.60 were deemed low quality, discarded, and subsequently repeated (Fig. 4B). In this way, we minimized the impact of erroneous data and statistical noise on virulence measurements.

## The time-to-death approach can be used to compare the virulence of *P. aeruginosa* strains

We next examined the utility of using the optimized $LT_{50}$ approach to compare the virulence of different *P. aeruginosa* strains. In addition to the bloodstream isolates PABL051 and PABL089, which were used in the preceding experiments, we tested four other bloodstream isolates: PABL038, PABL066, PABL083, and PABL093. These six isolates were chosen because they had previously been tested in a mouse model of bacteremia and shown to differ in their virulence (17, 18). The $LT_{50}$ methodology was applied to each of the six isolates, and $LT_{50}$ vs ln(dose) curves were generated (Fig. 5, which includes the PABL089 data shown in Fig. 1). Inspection of these curves suggested that lower bacterial doses better discriminated between strain virulence than higher doses. For

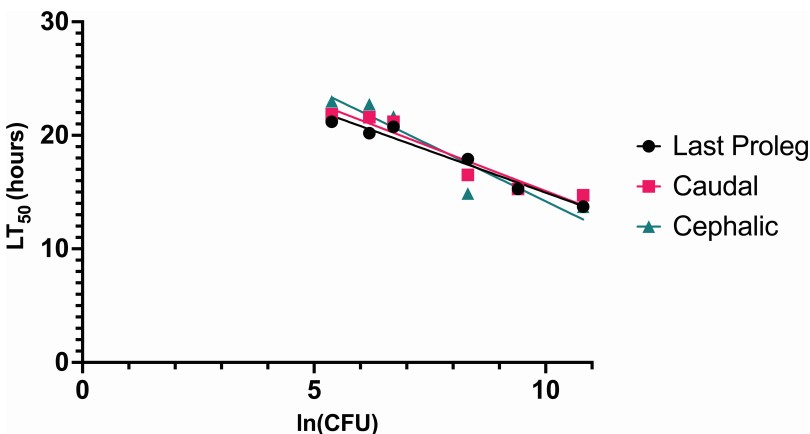

**FIG 3** Effect of injection site on LT$_{50}$ values following inoculation with *P. aeruginosa*. *P. aeruginosa* strain PABL089 was injected ~1 mm cephalic ("cephalic"), directly at ("Last Proleg") or ~1 mm caudal ("Caudal") to the larval final proleg, and LT$_{50}$ values were measured. Each symbol represents the LT$_{50}$ value associated with 10 larvae. The experiment was performed twice, and the results were combined. Regression lines were fit to the data and are shown. The differences between the three injection sites were not statistically significant. For the proleg, caudal, and cephalic injection groups, 95% CIs for LT$_{50}$(2,000 CFU) were 17.69–19.26, 17.67–20.02, and 17.11–20.78 hours, respectively; 95% CIs for LT$_{50}$(60,000 CFU) were 11.95–15.02, 11.23–15.82, and 8.63–15.78 hours, respectively.

example, at a dose of 2,000 CFU, the LT$_{50}$ values of the six strains varied from 12.6 to 18.2 hours, with PABL089 and PABL066 having statistically significantly higher LT$_{50}$ values and PABL083 having a statistically significantly lower LT$_{50}$ value than some of the other strains (Table 1 and Fig. 5; statistically significant differences are noted in the legend of Fig. 5). In contrast, at a theoretical dose of 60,000 CFU, the LT$_{50}$ values of the six strains ranged from 8.4 to 11.8 hours, with only PABL066 having an LT$_{50}$ value statistically significantly different from some of the other strains (Table 1 and Fig. 5; statistically significant differences are noted in the legend of Fig. 5). These findings indicate that a dose of 2,000 CFU better discriminated the virulence differences of these six strains than a dose of 60,000 CFU. For subsequent virulence comparisons, we used a dose of 2,000 CFU.

## *P. aeruginosa* strain virulence rankings in *G. mellonella* show modest agreement with those obtained using a mouse model of bacteremia

We compared the LT$_{50}$ virulence rankings of the six strains using *G. mellonella* to those previously reported using a mouse bacteremia model (Table 1). There was a modest correlation between mouse log(LD$_{50}$) values and *G. mellonella* LT$_{50}$(2,000 CFU): $R^2$ of 0.46 (Fig. 6). This finding confirms other reports suggesting that *G. mellonella* is a moderately useful surrogate for mice in predicting the virulence potential of *P. aeruginosa* (12). However, to our knowledge, this is the first investigation into the correlation between the *G. mellonella* model and the murine model for genetically diverse clinical *P. aeruginosa* isolates.

## In *G. mellonella,* the time-to-death approach has advantages over the LD$_{50}$ approach in measuring the virulence of *P. aeruginosa*

We have mentioned several advantages of the LT$_{50}$ approach over the conventional LD$_{50}$ approach. To illustrate this, we applied the LD$_{50}$ approach to measure the virulence of PABL038, a *P. aeruginosa* strain with a moderate level of virulence in both *G. mellonella* and mice (Table 1). *G. mellonella* larvae were infected with *P. aeruginosa* doses ranging from 22 to 580 CFU, and mortality was measured at 18- and 24-hours post-inoculation. Even when bacterial numbers as low as ~22 CFU were injected, all or nearly all the larvae died, making it difficult to estimate LD$_{50}$ values (Table 2). In contrast, the LT$_{50}$ approach

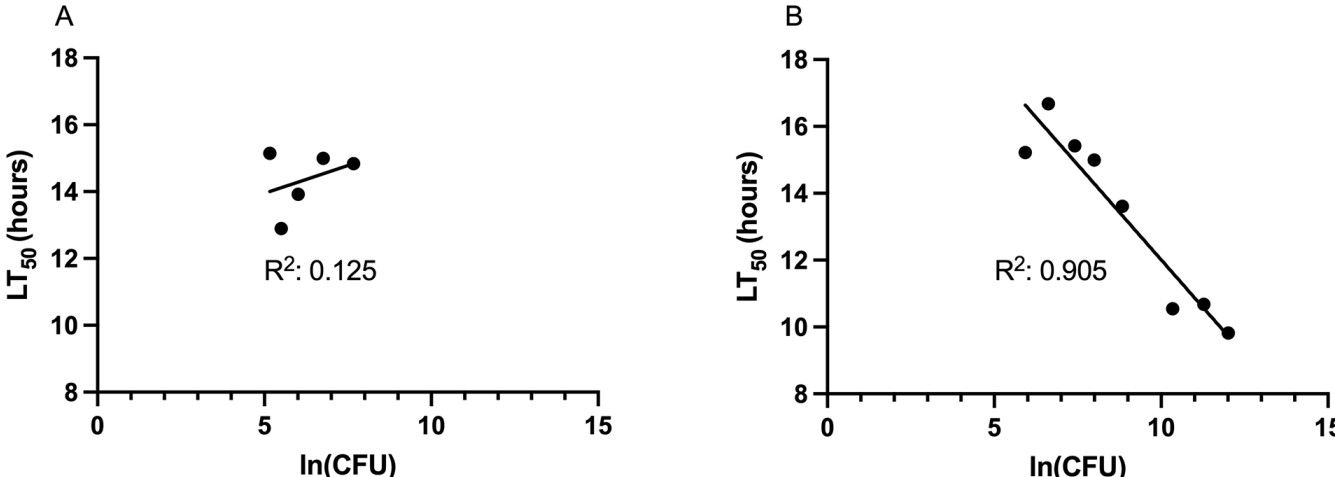

**FIG 4** Use of $R^2$ as a quality control metric. Different doses of strain PABL051 were injected into larvae, and the resulting LT$_{50}$ values were calculated. Each symbol represents 10 larvae. (A) In the initial experiment, data points fit poorly to a regression line ($R^2$ of 0.125), suggesting substantial error in the measurements. These data were discarded. (B) The experiment was repeated, and subsequent results showed much better alignment to the expected linear relationship ($R^2$ of 0.91). The second data set was deemed sufficiently accurate.

successfully captured the virulence of this strain (Table 1). This example illustrates the utility of the LT$_{50}$ approach in measuring the virulence of *P. aeruginosa*.

## DISCUSSION

*G. mellonella* is frequently used as a model organism for measuring bacterial virulence and response to antibiotics, but the highly pathogenic nature of *P. aeruginosa* toward *G. mellonella* poses special challenges. We provide an optimized protocol for the use of time-to-death as a virulence measure for *P. aeruginosa* infections of these larvae. We evaluated the impact of variables such as weight and injection site on this virulence readout. We also optimized an analytical approach that uses these results to allow for strain-to-strain comparisons even when relatively large uncertainties exist in dose estimations. Finally, we provide a quality control metric that is useful in ensuring that only reliable data are included in the final analysis. This protocol was tested on six *P. aeruginosa* clinical isolates and yielded results with moderate agreement to those previously obtained using a mouse model. This protocol may be useful to the *P. aeruginosa* research community.

Much of the previous work testing *P. aeruginosa* in *G. mellonella* has used LD$_{50}$ as a measure of virulence (11, 12, 15, 23–25). Many of these reports compared parental strains of *P. aeruginosa* to mutant strains that were defective in the production of critical pathogenic factors and were, therefore, highly attenuated in virulence. The LD$_{50}$ approach is well suited for such comparisons. However, we found it performed suboptimally when comparing strains that had only small differences in virulence. For example, the strains we investigated killed nearly all larvae over a period of 18–24 hours even with bacterial doses as low as 22 CFU (Table 2), making it difficult to determine an LD$_{50}$ value. In contrast, the LT$_{50}$ approach distinguished virulence differences between some of these strains. A second approach favored by investigators testing the efficacy of antimicrobial compounds is to compare Kaplan-Meier survival curves of infected *G. mellonella* larvae exposed to antibiotics or vehicles (26–28). This focus on survival differences between treatment and non-treatment cohorts circumvents the issues associated with measuring LD$_{50}$ values. However, this approach (which in essence is an LT$_{50}$ measurement for a single dose) works less well when comparing the survival of multiple *P. aeruginosa* strains because it requires that the same inoculum of each strain be given to the cohorts of larvae. It becomes difficult to determine whether

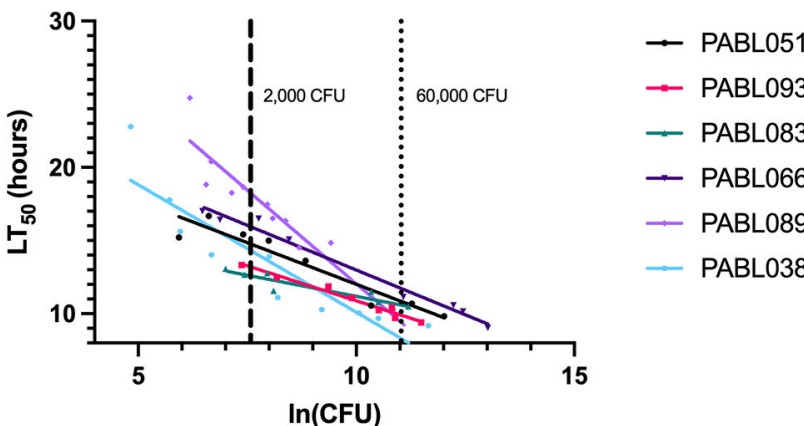

**FIG 5** Virulence of six strains of *P. aeruginosa* in *G. mellonella*. LT$_{50}$ vs ln(dose) data were generated for six bloodstream isolates of *P. aeruginosa*. Regression lines were then fit to each set of data points. Each data point represents the results of a single experiment in which 10 larvae received a single dose of bacteria. Each experiment was performed using at least five different bacterial doses. Each experiment consists of at least two biological replicates performed by different operators. Cumulative results are displayed. At a dose of 2,000 CFU, statistically significant differences (non-overlapping 95% CIs) were observed between PABL089 and PABL051, PABL066, PABL083, PABL093, PABL038; between PABL066 and PABL083, PABL093; between PABL083 and PABL051; between PABL093 and PABL051. At a dose of 60,000 CFU, statistically significant differences were observed between PABL066 and PABL093, PABL083, PABL038.

measured differences in virulence between two strains are biological or are instead due to small differences in the inoculum sizes or to the experimental uncertainties in the measurements. The LT$_{50}$ approach detailed here accurately captures small differences in virulence between even highly pathogenic strains by using multiple measurements and allowing for interpolation of time-to-death for any inoculum size.

A comparison of the virulence of different strains unexpectedly uncovered an interesting aspect of *P. aeruginosa* virulence. We tested six *P. aeruginosa* bloodstream isolates previously shown to have a wide range of virulence in a mouse model of bacteremia (17, 18). The LT$_{50}$ values of these six strains were more distinct at low inoculums (~2,000 CFU) than at high inoculums (~60,000 CFU). In other words, at a high inoculum, all six strains killed *G. mellonella* at about the same time, whereas at a low inoculum, they differed somewhat in the time required to cause death. What strain differences, such as quorum sensing, biofilm formation, or growth rate, account for these dose-dependent effects is an interesting avenue of future study and may inform *P. aeruginosa* pathogenesis.

A critical measure of the usefulness of the *G. mellonella* model for the study of *P. aeruginosa* is how closely virulence measurements in *G. mellonella* reflect those obtained in mice. Although *G. mellonella* lacks an adaptive immune response, this organism's innate immune system has similarities with those of vertebrates (2). Therefore, it has been proposed that *G. mellonella* may be quite useful for modeling acute *P. aeruginosa*

**TABLE 1** Virulence measurements of *P. aeruginosa* bloodstream isolates in *G. mellonella* and mice

| Strain name | *G. mellonella* LT$_{50}$ at 2,000 CFU (hours) | *G. mellonella* LT$_{50}$ at 60,000 CFU (hours) | Murine LD$_{50}$ (CFU)[a] |
|---|---|---|---|
| PABL083 | 12.6 | 10.6 | 6.4 |
| PABL093 | 13.2 | 9.9 | 6.3 |
| PABL038 | 14.3 | 8.4 | 7.4 |
| PABL051 | 14.7 | 10.9 | 7.5 |
| PABL066 | 15.9 | 11.8 | 7.7 |
| PABL089 | 18.2 | 9.5 | 7.3 |

[a]Mouse LD$_{50}$ values were previously reported (17, 18).

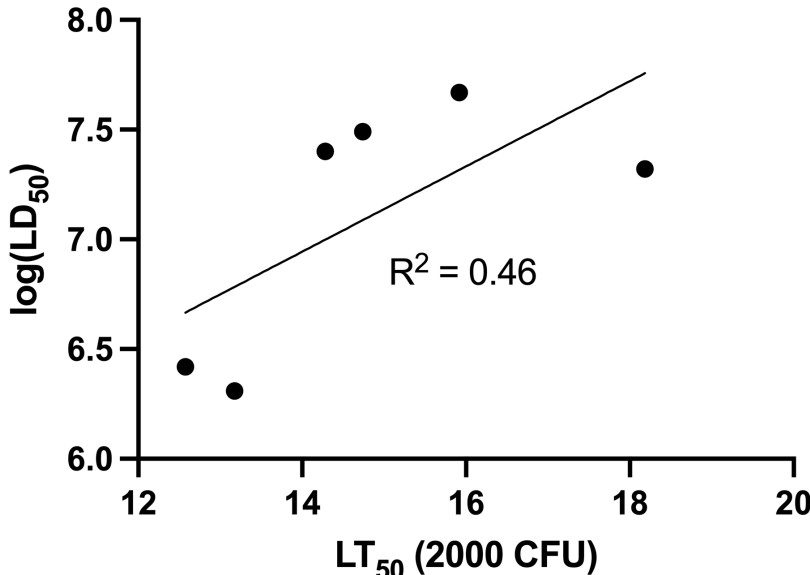

$R^2 = 0.46$

**FIG 6** Comparison of the virulence of six *P. aeruginosa* strains in the *G. mellonella* model and the murine model. Virulence measures in the *G. mellonella* model ($LT_{50}$ values at a dose of 2,000 CFU) and a murine model of bacteremia ($LD_{50}$ values) were plotted, and a moderate level of agreement was observed ($R^2 = 0.46$).

infections, in which the innate immune response is critical (29). Results from the *G. mellonella* model were moderately similar to those from mice (Table 1). Although these observations are intriguing, studies using many more *P. aeruginosa* strains obtained from a variety of clinical and environmental sources will be necessary to demonstrate the true usefulness and limitations of the *G. mellonella* model. Similarly, the degree of agreement between *G. mellonella* and mice may also depend on the type (breed, immunocompromised status, and site of infection) of the mouse model used.

Our study has several limitations. As mentioned above, only bloodstream isolates of *P. aeruginosa* were used, and other types of isolates (pneumonia, urinary tract infections, and natural environment) will need to be tested to show the generalizability of the approach. In particular, the use of cystic fibrosis isolates with this $LT_{50}$ methodology will be important, as these isolates differ markedly in their virulence compared to acute infection isolates (25). We applied the protocol to only six strains. A much larger collection of strains will need to be tested to more definitively determine how well this *G. mellonella* model ranks *P. aeruginosa* strain virulence compared to mice. Finally, we purchased *G. mellonella* larvae from a non-scientific commercial vendor, and the genetics and breeding history of these larvae are unclear. We cannot be certain that other groups using *G. mellonella* from different sources will obtain results similar to ours. The purchase of inbred *G. mellonella,* which are currently available in Europe (30) but not (to our knowledge) in the United States, would circumvent this issue.

In summary, we provide a protocol for reproducibly measuring and comparing the virulence of *P. aeruginosa* using *G. mellonella*. This approach may be useful in increasing our understanding of *P. aeruginosa* pathogenesis and the mechanisms by which this bacterium causes disease. It may also be applicable to other bacterial species, especially

**TABLE 2** Number of dead larvae following injection of different-sized inoculums of strain PABL038[a]

| Hours post-infection | Inoculum size | | | | |
|---|---|---|---|---|---|
| | 580 CFU | 265 CFU | 160 CFU | 40 CFU | 22 CFU |
| 18 | 10 | 10 | 9 | 9 | 10 |
| 24 | 10 | 10 | 9 | 9 | 10 |

[a]Total of 10 larvae were injected for each group.

those that are highly virulent in *G. mellonella* (e.g., *Burkholderia pseudomallei* and *Burkholderia mallei* [31]) and for which the lethal dose methodology is not optimal. In addition, this methodology should be applicable to efficacy studies for testing antibiotics and alternative therapeutics.

## ACKNOWLEDGMENTS

Support for this work was provided by the National Institutes of Health awards RO1 AI118257, K24 AI04831, R21 AI129167, R21 AI153953 (all to A.R.H.), and T32 AI007476 (T.K.); by the American Cancer Society Postdoctoral Fellowship (130602-PF-17-107-01-MPC, awarded to K.E.R.B.); by Northwestern University Undergraduate and Academic Year Research Grants; and by the Northwestern University Bioscientist Program. Statistical services were provided by the Northwestern University Biostatistics Collaboration Center.

## AUTHOR AFFILIATIONS

[1]Department of Microbiology-Immunology, Feinberg School of Medicine, Northwestern University, Chicago, Illinois, USA
[2]Northwestern University, Evanston, Illinois, USA
[3]Yale University, New Haven, Connecticut, USA
[4]Northeastern Illinois University, Chicago, Illinois, USA
[5]Tufts University, Medford, Massachusetts, USA
[6]Hamilton College, Clinton, New York, USA
[7]Illinois Mathematics and Science Academy, Aurora, Illinois, USA
[8]Department of Pharmacy Practice, Midwestern University Colleges of Pharmacy and Pharmacology, Downers Grove, Illinois, USA
[9]Pharmacometrics Center of Excellence, Midwestern University, Downers Grove, Illinois, USA
[10]Division of Infectious Diseases, Department of Medicine, Feinberg School of Medicine, Northwestern University, Chicago, Illinois, USA

## PRESENT ADDRESS

Travis J. Kochan, Laboratory of Respiratory and Special Pathogens, Division of Bacterial, Parasitic, and Allergenic Products, Office of Vaccines Research and Review, Center for Biologics, Evaluation and Research, Food and Drug Administration, Silver Spring, Maryland, USA
Timothy L. Turner, Abbott Laboratories, Abbott Park, Illinois, USA
Nathan B. Pincus, Department of Medicine, Stanford University, Stanford, California, USA

## AUTHOR ORCIDs

Christopher M. R. Axline http://orcid.org/0009-0002-8392-5991
Timothy L. Turner http://orcid.org/0000-0003-4276-5864
Marc H. Scheetz http://orcid.org/0000-0002-1091-6130
Kelly E. R. Bachta https://orcid.org/0000-0001-8838-5117
Alan R. Hauser http://orcid.org/0000-0003-4596-7939

## AUTHOR CONTRIBUTIONS

Christopher M. R. Axline, Data curation, Formal analysis, Investigation, Methodology, Software, Supervision, Validation, Visualization, Writing – original draft, Writing – review and editing | Travis J. Kochan, Conceptualization, Data curation, Formal analysis, Investigation, Methodology, Project administration, Software, Supervision, Validation, Visualization, Writing – review and editing | Sophie Nozick, Data curation, Formal analysis, Investigation, Methodology, Project administration, Supervision, Validation, Visualization, Writing – review and editing | Timothy Ward, Conceptualization, Formal

analysis, Investigation, Methodology, Software, Supervision, Validation, Visualization, Writing – review and editing | Tania Afzal, Conceptualization, Data curation, Formal analysis, Investigation, Methodology, Supervision, Validation, Visualization, Writing – review and editing | Issay Niki, Data curation, Formal analysis, Investigation, Methodology, Supervision, Validation, Visualization, Writing – review and editing | Sumitra D. Mitra, Conceptualization, Data curation, Formal analysis, Investigation, Methodology, Supervision, Validation, Visualization, Writing – review and editing | Ethan VanGosen, Data curation, Formal analysis, Investigation, Methodology, Validation, Visualization, Writing – review and editing | Julia Nelson, Data curation, Formal analysis, Investigation, Methodology, Validation, Visualization, Writing – review and editing | Aliki Valdes, Formal analysis, Investigation, Methodology, Validation, Visualization, Writing – review and editing | David Hynes, Formal analysis, Investigation, Methodology, Validation, Visualization, Writing – review and editing | William Cheng, Formal analysis, Investigation, Methodology, Validation, Visualization, Writing – review and editing | Joanne Lee, Formal analysis, Investigation, Methodology, Validation, Visualization, Writing – review and editing | Prarthana Prashanth, Formal analysis, Investigation, Methodology, Validation, Visualization, Writing – review and editing | Timothy L. Turner, Conceptualization, Data curation, Formal analysis, Investigation, Methodology, Supervision, Validation, Visualization, Writing – review and editing | Nathan B. Pincus, Conceptualization, Formal analysis, Investigation, Methodology, Supervision, Validation, Visualization, Writing – review and editing | Marc H. Scheetz, Conceptualization, Writing – review and editing | Kelly E. R. Bachta, Conceptualization, Data curation, Formal analysis, Investigation, Methodology, Project administration, Supervision, Validation, Visualization, Writing – review and editing | Alan R. Hauser, Conceptualization, Formal analysis, Funding acquisition, Methodology, Project administration, Resources, Supervision, Writing – review and editing

## ADDITIONAL FILES

The following material is available online.

### Supplemental Material

**Supplemental figures (Spectrum01666-24-s0001.pdf).** Fig. S1 and S2.
**Supplemental methods (Spectrum01666-24-s0002.pdf).** Protocol for performing *Galleria* experiments.

### Open Peer Review

**PEER REVIEW HISTORY (review-history.pdf).** An accounting of the reviewer comments and feedback.

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
