## [Reviewer comments · Microbiology Spectrum]

Microbiology Spectrum

Refined methodology for quantifying *Pseudomonas aeruginosa* virulence using *Galleria mellonella*

Christopher Axline, Travis Kochan, Sophie Nozick, Timothy Ward, Tania Afzal, Issay Niki, Sumitra Mitra, Ethan VanGosen, Julia Nelson, Aiki Valdes, David Hynes, William Cheng, Joanne Lee, Prarthana Prashanth, Timothy Turner, Nathan Pincus, Marc Scheetz, Kelly Bachta, and Alan Hauser

Corresponding Author(s): Alan Hauser, Northwestern University Feinberg School of Medicine

Review Timeline:

Submission Date:	July 8, 2024
Editorial Decision:	July 29, 2024
Revision Received:	October 14, 2024
Accepted:	November 15, 2024

Editor: Mariola Ferraro

Reviewer(s): Disclosure of reviewer identity is with reference to reviewer comments included in decision letter(s). The following individuals involved in review of your submission have agreed to reveal their identity: Alyssa Walker (Reviewer #1)

Transaction Report:

DOI: <https://doi.org/10.1128/spectrum.01666-24>

Re: Spectrum01666-24 (Refined methodology for quantifying *Pseudomonas aeruginosa* virulence using *Galleria mellonella*)

Dear Dr. Alan R. Hauser:

Thank you for the privilege of reviewing your work. Below you will find my comments, instructions from the Spectrum editorial office, and the reviewer comments.

Revision Guidelines

Sincerely,
Mariola Ferraro
Editor
Microbiology Spectrum

Reviewer #2 (Comments for the Author):

Summary:

Axline et al. put forward standardized methodology for *G. mellonella* *P. aeruginosa* infections. They control for multiple factors: injection location, larval weight, and take into account the actual CFU delivered. They provide an available R script to automate calculation of the timepoint where 50% of larvae have died, and quality control checks that use several inoculum concentrations.

With some clarifications, this will be a valuable resource for the *P. aeruginosa* research community.

Comments:

1. Figure 1 may need to be larger, and it would be helpful to show the injection point here. Do the researchers intend to aim precisely for either the left or right proleg? Being unfamiliar with injecting insect larvae, is the proleg a soft structure that can be injected easily? The supplement figure S3 shows an arrow pointed slightly adjacent to the proleg. Is there a recommended angle to hold the larva? It is difficult to determine from the small photo and top-down view.
2. The use of highly virulent clinical isolates is a reasonable choice for this study. Figure 4 uses lab strain PAO1. Is it feasible to include PAO1 in Figure 6? It would be interesting to directly compare if lab strains have similar LT50 values to clinical isolates.
3. On line 270, Usage of comparing antimicrobial compounds is mentioned. How are antibiotic compounds or other anti-virulence drugs administered to *G. mellonella*, and how would this model be used to study these types of effects?
4. Line 290 and following, discusses factors that may contribute to varying slopes of the lines, and speculates these may differ between *P. aeruginosa* strains. Is there existing literature using *G. mellonella* studies that supports any of these possibilities?

Minor comments

1. Line 56, assume "wax" is intended.
2. Figure 2A requires higher resolution.

Summary

The authors propose an alternative methodology, LT50, for assessing the virulence of *P. aeruginosa* in *G. mellonella*, addressing challenges posed by the traditional LD50 method. They also present an analytical framework for comparing the virulence of different *P. aeruginosa* strains using LT50 values, thereby establishing a standardized approach for evaluating bacterial pathogenicity in the *G. mellonella* model. Overall, this study contributes a standardized methodology that enhances the accuracy and reproducibility of virulence assessments using *G. mellonella* larvae as a model, thereby advancing research in microbial pathogenesis through comparative studies of the virulence of various bacterial strains. However, several issues need addressing. Of particular concern is the low number of biological replicates and independent experiments. Additionally, the authors occasionally omit clear statements regarding the significance or broader implications of their findings, and some sections of the results require readers to infer meaning rather than explicitly stating it.

Major revisions:

I'm concerned about the safety of the injection method used for the worms. Some studies use a sponge and clamp to secure the worm, which prevents the researcher from accidentally inoculating himself. Given the virulence of clinical *P. aeruginosa* samples, likely containing multiple antimicrobial-resistant genes, the authors should emphasize safety and precautions. They should, at minimum, cite studies using the sponge method (which I've used; it takes a few minutes to adjust but works well) to advertise the use of extra precautions. Additionally, it's common to anesthetize *Galleria larvae* with ice to immobilize them and prevent accidental needle sticks.

Reference: Dalton JP, Uy B, Swift S and Wiles S (2017) A Novel Restraint Device for Injection of *Galleria mellonella* Larvae that Minimizes the Risk of Accidental Operator Needle Stick Injury. *Front. Cell. Infect. Microbiol.* 7:99. doi: 10.3389/fcimb.2017.00099

The first paragraph of the results section lacks supporting data, crucial as it signifies the authors' standardized method and critiques inferior approaches. Data must be shown to substantiate these claims. As an example, refer to the study (referenced below) where results obtained by using traditional methods were used to assert superiority of the proposed approach. Reference: Walker AC, Bhargava R, Vaziriyani-Sani AS, Brust AS, Czyz DM. Quantification of Bacterial Loads in *Caenorhabditis elegans*. *Bio Protoc.* 2022 Jan 20;12(2):e4291. doi: 10.21769/BioProtoc.4291. PMID: 35127981; PMCID: PMC8799673.

Lines 151-153 does not make sense. If one is comparing the LT50 of two strains at a certain inoculum, why don't they need to prepare suspensions of exactly 2000 CFU for each strain? Even if that is the case, how is that different from what is required for LD50? Please clarify.

Authors do not state the conclusions or significance from the figures referenced in lines 155-163. At minimum, the point/significance of these figures should be mentioned collectively at the end of this paragraph.

Figures 3,4: it is concerning that only 10 worms were used. How many individual experiments? Authors should indicate the number of individual experiments, and if only one, please repeat. Additionally, there are conclusions made in the manuscript about the data presented but there are no statistics performed on these data. Are regression lines significantly different from one another? Authors need to run statistics and insert in the text appropriately.

Lines 178-187: Authors should include a figure indicating exact injection site relative to the proleg. They mention (~1 mm); was this measurement precise? Injection into the body cavity

can be fatal, especially with needles larger than 30-31G. The proleg's thicker cuticle protects against the mechanical intrusion of the needlestick, and the hemolymphs (insect blood) in the proleg facilitates distribution.

Authors should show AUC graphs in addition to the table with the values.

In table 1, the authors include AUC data from a mouse experiment in another paper, but fail to mention the reason for doing so in this section. I believe lines 306-308 could be moved to this result section to resolve this issue.

Figure 6 is only one individual experiment. Because this publication claims to increase reproducibility, the authors should replicate the experiment done in Figure 6 (which was only done once) and show the two independent experiments side-by-side. Repeating this experiment is also important because it is further discussed in the discussion.

Figure S3 needs to be modified such that the arrow is pointing to the inner circle in the proleg (assuming that is what the researchers did). The arrow placement suggests that the researcher injected into the body cavity (which can increase the risk of death).

Lines 321-323: the authors say they used worms that were not inbred- how do they know this? I did not see in the methods that authors cultured their own colony. The reference provided in the text is not sufficient. Unless this is corrected authors should remove these sentences.

Minor revisions:

The importance section could be improved by explicitly stating the importance(s) of the present publication.

Lines 49-50: "Virulence measurements of clinical isolates are typically performed in murine models, which possess inherent limitations such as cost, labor, and ethical concerns." There are an abundance of studies that use tissue culture and invertebrate (*C. elegans*, *Drosophila*) models to look at this, which do not have these limitations. Authors should re-word that sentence appropriately so that it is not misleading.

Lines 60,69-71, 303-304: needs references. Line 167: references are improperly inserted- there needs to be a comma after (20) or they both need to be placed in parenthesis, together, at the end of the sentence.

I am curious if the authors repeated any of these experiments with a smaller gauge needle- I have found that insulin needles (30-31G) induce far less trauma to the worms than larger gauges. Because this is a methods paper, it seems like a missed opportunity to not include this, but this is a minor comment and in my opinion not required for publication.

The authors need to list RPM of shaking incubator in which bacteria are cultured (line 108) and in the supplemental methods.

The authors' discussion of CFU/mL is confusing. While I now understand from reading the entire manuscript that CFU/mL was used to determine the dose, the method suggested otherwise in line 110, causing confusion. Throughout the manuscript, the authors should clarify that CFU/mL is used for each strain. For instance, in line 156, where larvae were infected with an "estimated dose," the authors should explicitly clarify this and provide the actual dose in the manuscript.

It is unusual that Figure 1 is referenced only in the methods and not the results section, but this is up to the handling editor.

In the methods section, the authors mention that at least two biological replicates were conducted, each performed by a different researcher. I believe they intend to refer to two independent experiments or technical replicates, rather than two individual worms (1 worm = 1 biological replicate). Please clarify.

Figure 4: shows 1) caudal to proleg, 2) cephalic to proleg, 3) proleg, but in the text the authors only mention caudal and cephalic to proleg; the authors also need to mention in the text that the proleg was also injected. Additionally, because this is a methods paper, the authors should mention why the proleg is commonly as the injection site.

The authors should re-think the conclusion made in lines 226-227 that their findings highlight the "complex" relationship between..., and if the authors choose to keep it, they should expand on how exactly their findings highlight a complex relationship.

The discussion repeatedly talks about how the present manuscript is important for *P. aeruginosa* research but fail to mention a broader application. Authors should not limit themselves to *P. aeruginosa* in the discussion and should mention the potential for its application in studies of other virulent bacteria beyond *P. aeruginosa*, particularly in the concluding paragraph of the discussion.

The authors say that smaller AUCs reflect more virulent strains- is this an assumption based on death of the larvae? Is there known virulence of their clinical isolates samples that correlate to their results? Is it because of the mouse study? Please clarify.

Lines 321-323: authors use "inbred" and "in-bred"- be consistent.

Galleria are usually kept in a refrigerator to prevent metamorphosis. If this is the case for the present paper, authors should discuss how long they leave them out of the refrigerator before they assess movement for larval sorting in step 1 of supplemental methods.

The supplemental methods section describes how to dilute sample to obtain CFU/mL but does not mention that the researcher has to plate the dilutions and count colonies. Please add.

In the supplemental methods, the authors should explain the reason for various steps. One example is "ensure that the bevel of the needle is facing upwards when..."- why?

In the supplemental methods, the statement "16. Note that..." requires a reference. Additionally, worms may appear immobile when shaking the dish, but still exhibit mouth movements observable under a dissection microscope. Therefore, the authors should specify that *they* designated larvae as dead when no movement was observed after shaking, to prevent confusion that no movement unequivocally means they are dead.

Response to Reviewers

We thank the reviewers for their thoughtful and constructive criticisms. Below we address each in turn.

Reviewer 1

Major

1. *I'm concerned about the safety of the injection method used for the worms. Some studies use a sponge and clamp to secure the worm, which prevents the researcher from accidentally inoculating himself. Given the virulence of clinical *P. aeruginosa* samples, likely containing multiple antimicrobial-resistant genes, the authors should emphasize safety and precautions. They should, at minimum, cite studies using the sponge method (which I've used; it takes a few minutes to adjust but works well) to advertise the use of extra precautions. Reference: Dalton JP, Uy B, Swift S and Wiles S (2017) A Novel Restraint Device for Injection of *Galleria mellonella* Larvae that Minimizes the Risk of Accidental Operator Needle Stick Injury. *Front. Cell. Infect. Microbiol.* 7:99. doi: 10.3389/fcimb.2017.00099. We thank the reviewer for this suggestion. We now list the sponge/clamp system in our protocol and cite this reference.*

2. *Additionally, it's common to anesthetize *Galleria* larvae with ice to immobilize them and prevent accidental needle sticks. We appreciate this suggestion but have anecdotal evidence that cooling the larvae prior to injection may alter their time to death. For this reason, we have not added this modification to our protocol. Of note, we have now taught six inexperienced undergraduate students this approach, and they have injected a minimum of 15,000 larvae without anesthetic or cooling, and we have not experienced a single needlestick.*

3. *The first paragraph of the results section lacks supporting data, crucial as it signifies the authors' standardized method and critiques inferior approaches. Data must be shown to substantiate these claims. As an example, refer to the study (referenced below) where results obtained by using traditional methods were used to assert superiority of the proposed approach. Reference: Walker AC, Bhargava R, Vaziriyani-Sani AS, Brust AS, Czyz DM. Quantification of Bacterial Loads in *Caenorhabditis elegans*. *Bio Protoc.* 2022 Jan 20;12(2):e4291. doi: 10.21769/BioProtoc.4291. PMID: 35127981; PMCID: PMC8799673. We have now added two figures (Supplemental Fig 1 and Supplemental Fig 2) to support the statements in this paragraph.*

4. *Lines 151-153 does not make sense. If one is comparing the LT50 of two strains at a certain inoculum, why don't they need to prepare suspensions of exactly 2000 CFU for each strain?* For the two reasons stated in the preceding paragraph of the manuscript, preparing an inoculum of exactly 2000 CFU for two different *P. aeruginosa* strains is difficult. As suggested by this reviewer, we have now added two figures (Supplemental Fig 1 and Supplemental Fig 2) to illustrate these difficulties. As a hypothetical example, suppose one attempted to inoculate 2000 CFU of strain 1 and strain 2 into larvae but actually inoculated 1,500 CFU of strain 1 and 2,500 CFU of strain 2. If strain 1 yields an LT50 value of 16 hours, and strain 2 yields an LT50 value of 14 hours, one will not be able to ascertain whether these differences were due to a true virulence disparity (i.e., strain 1 is less virulent than strain 2) or were due to the lower inoculum of strain 1 relative to strain 2. The use of LT50 vs. $\ln(\text{dose})$ curves circumvents this difficulty. The regression line fit to the LT50 vs. $\ln(\text{dose})$ data points generates an equation that provides LT50 values as a function of $\ln(\text{dose})$. The LT50 value corresponding to a dose of 2000 CFU

can be estimated from this equation even if the operator was unable to prepare an injection dose of exactly 2000 CFU. We have revised this section to more clearly make this point: “For example, the most straightforward method of comparing virulence would be to compare the LT_{50} values of the two strains at a specific inoculum (e.g., 2000 CFU). However, as described above, it can be quite difficult to prepare bacterial suspensions of exactly 2000 CFU for two different strains (Supplemental Fig 2). The regression line fit to the LT_{50} vs. $\ln(\text{dose})$ data points generates an equation that provides LT_{50} values as a function of $\ln(\text{dose})$. The LT_{50} value corresponding to a dose of 2000 CFU can be estimated from this equation even if the operator was unable to prepare an injection dose of exactly 2000 CFU.”

5. *Even if that is the case, how is that different from what is required for LD50? Please clarify.* The advantage of the LT_{50} methodology over the LD50 methodology is that many strains of *P. aeruginosa* kill all larvae regardless of the bacterial dose, making it impossible to calculate a LD50 value. Even if all larvae die following injection of a particular dose of bacteria, one can still calculate an LT_{50} value by measuring how quickly they die. The advantage of the LT_{50} methodology over the LD50 methodology is discussed (Table 2).

6. *Authors do not state the conclusions or significance from the figures referenced in lines 155-163. At minimum, the point/significance of these figures should be mentioned collectively at the end of this paragraph.* We have now added the following conclusion to this paragraph: “This example illustrates how the time-to-death methodology can be used to compare the virulence of *P. aeruginosa* strains.”

7. *Figures 3,4: it is concerning that only 10 worms were used. How many individual experiments? Authors should indicate the number of individual experiments, and if only one, please repeat.* Note that 10 worms were used per dose, and that each experiment consists of 6-8 doses, for a total of 60-80 worms per strain. We have now repeated these experiments such that each experiment has been performed twice. We have also redone the injection site experiment (the new Fig 3) using strain BL089 instead of strain PAO1 so that it is now consistent with the weight experiment (the new Fig 2).

8. *Additionally, there are conclusions made in the manuscript about the data presented but there are no statistics performed on these data. Are regression lines significantly different from one another? Authors need to run statistics and insert in the text appropriately.* We thank the reviewer for this suggestion. We have now consulted with our Statistics Core and have added a description of statistical analyses to the Methods section and state which data sets are statistically significantly different in the figure legends of the relevant figures.

9. *Lines 178-187: Authors should include a figure indicating exact injection site relative to the proleg.* We have revised this paragraph to direct the reader to a photograph of a larva in the Supplemental Methods, in which the exact injection site is indicated. “In previous studies, *G. mellonella* were most commonly injected at the final proleg (see the Supplemental Methods section for photograph) (21, 22). We examined whether injections with *P. aeruginosa* strain PAO1 slightly (~1 mm) cephalic or caudal to the final proleg or at the site of the proleg altered LT_{50} values.”

10. *They mention (~1 mm); was this measurement precise? Injection into the body cavity can be fatal, especially with needles larger than 30-31G. The proleg's thicker cuticle protects against the mechanical intrusion of the needlestick, and the hemolymphs (insect blood) in the proleg*

facilitates distribution. The injection sites were approximate. The purpose of this experiment was not to define the impact of a precise injection site on mortality but rather to determine whether injection site was a variable that could impact *Galleria* results and therefore needed to be carefully monitored. In this experiment, we wished to mimic the slight variability in injection site that may occur between operators or when a new operator is learning the technique. For this reason, we did not define an exact injection site 1 mm cephalic or caudal to the proleg but rather had the operator estimate such a site. From these experiments, we concluded that small variations in injection site did not alter time to death in *Galleria*, and a focus on precisely defining injection site would not decrease the variability observed in time-to-death measurements. We have now revised this section to emphasize that the injection sites were approximate: “We examined whether injections with *P. aeruginosa* strain PAO1 slightly (approximately 1 mm) cephalic or caudal to the final proleg or at the site of the proleg altered LT₅₀ values.”

11. *Authors should show AUC graphs in addition to the table with the values.* In the process of addressing point 8 above, we have removed the AUC data and comparisons from the manuscript.

12. *In table 1, the authors include AUC data from a mouse experiment in another paper, but fail to mention the reason for doing so in this section. I believe lines 306-308 could be moved to this result section to resolve this issue.* We assume the reviewer is referring to the Results section entitled, “The time-to-death approach can be used to compare the virulence of *P. aeruginosa* strains,” as this is where Table 1 is first cited. We have now added the following sentence to this section: “These six isolates were chosen because they had previously been tested in a mouse model of bacteremia and shown to differ in their virulence (17, 30).”

13. *Figure 6 is only one individual experiment. Because this publication claims to increase reproducibility, the authors should replicate the experiment done in Figure 6 (which was only done once) and show the two independent experiments side-by-side. Repeating this experiment is also important because it is further discussed in the discussion.* We neglected to mention in the legend that these experiments were each performed twice, and the cumulative results displayed. We have now revised the figure legend to make this clear: “Each data point represents the results of a single experiment in which 10 larvae received a single dose of bacteria. Each experiment was performed using at least five different bacterial doses and repeated. Cumulative results are shown.” Below, we show the results of the individual experiments.

14. Figure S3 needs to be modified such that the arrow is pointing to the inner circle in the proleg (assuming that is what the researchers did). The arrow placement suggests that the researcher injected into the body cavity (which can increase the risk of death). Injections were indeed performed in the inner circle of the proleg. We have modified this figure (now Supplemental Fig 4) to more clearly show this.

15. Lines 321-323: the authors say they used worms that were not inbred- how do they know this? I did not see in the methods that authors cultured their own colony. The reference provided in the text is not sufficient. Unless this is corrected authors should remove these sentences. This is a good point. We have now revised this sentence: “Finally, we purchased *G. mellonella* larvae from a non-scientific commercial vendor, and the genetics and breeding history of these larvae are unclear. We cannot be certain that other groups using *G. mellonella* from different sources will obtain results similar to ours. Purchase of in-bred *G. mellonella*, which are currently available in Europe (31) but not (to our knowledge) in the United States, would circumvent this issue.”

Minor revisions:

16. The importance section could be improved by explicitly stating the importance(s) of the present publication. We have revised the Importance section as follows: “*Pseudomonas aeruginosa* is a significant cause of morbidity and mortality. The invertebrate *Galleria mellonella* is used as a model to determine the virulence of *P. aeruginosa* strains. We provide a protocol and analytical approach for using a time-to-death metric to accurately compare the virulence of *P. aeruginosa* strains in *G. mellonella* larvae. This methodology, which has several advantages over 50% lethal dose approaches, is a useful resource for the study of *P. aeruginosa* pathogenicity.”

17. Lines 49-50: “Virulence measurements of clinical isolates are typically performed in murine models, which possess inherent limitations such as cost, labor, and ethical concerns.” There are an abundance of studies that use tissue culture and invertebrate (*C. elegans*, *Drosophila*) models to look at this, which do not have these limitations. Authors should re-word that sentence appropriately so that it is not misleading. In rewriting the Importance section, we have removed this sentence.

18. Lines 60,69-71, 303-304: needs references. We have added references.

19. Line 167: references are improperly inserted- there needs to be a comma after (20) or they both need to be placed in parenthesis, together, at the end of the sentence. We have added a comma.

20. I am curious if the authors repeated any of these experiments with a smaller gauge needle- I have found that insulin needles (30-31G) induce far less trauma to the worms than larger gauges. Because this is a methods paper, it seems like a missed opportunity to not include this, but this is a minor comment and in my opinion not required for publication. We have not used smaller needles but thank the reviewer for this suggestion. We will explore this in our future work.

21. The authors need to list RPM of shaking incubator in which bacteria are cultured (line 108) and in the supplemental methods. We have added this information.

22. The authors' discussion of CFU/mL is confusing. While I now understand from reading the entire manuscript that CFU/mL was used to determine the dose, the method suggested otherwise in line 110, causing confusion. Throughout the manuscript, the authors should clarify that CFU/mL is used for each strain. We now refer the reader to the Supplemental Methods for a detailed description of how dosing for all experiments was measured and also clarify the sentence identified by the reviewer as follows: "A detailed experimental protocol describing the *G. mellonella* methodology is provided in the Supplemental Methods. Briefly, isolates were grown overnight at 37°C in Lysogeny broth (LB) with shaking (250 RPM) and then sub-cultured in LB, pelleted, and resuspended in phosphate-buffered saline (PBS). Estimated doses for injection into larvae were obtained by diluting bacteria in PBS to an optical density at 600 nm (OD₆₀₀) of ~0.2 (the equivalent of ~5 × 10⁷ CFU/ml) and making appropriate dilutions with PBS to generate a range of bacterial doses (i.e., a range of suspensions of different CFU/ml) for injection into *G. mellonella*."

23. For instance, in line 156, where larvae were infected with an "estimated dose," the authors should explicitly clarify this and provide the actual dose in the manuscript. We have revised this sentence as follows: "Sets of ten larvae were infected with a range of doses of PABL089 (485, 705, 790, 1265, 1650, 2867, 3250, 4350, 5950, 12300 CFU), and larval death was monitored each hour."

24. It is unusual that Figure 1 is referenced only in the methods and not the results section, but this is up to the handling editor. We have removed this figure.

25. In the methods section, the authors mention that at least two biological replicates were conducted, each performed by a different researcher. I believe they intend to refer to two independent experiments or technical replicates, rather than two individual worms (1 worm = 1 biological replicate). Please clarify. We have added to the Methods section "Each inoculum for each *P. aeruginosa* isolate was injected into 10 larvae (i.e., 10 technical replicates)." We have removed the sentence about biological replicates from the Methods section and now address the number of replicates in each figure legend.

26. Figure 4: shows 1) caudal to proleg, 2) cephalic to proleg, 3) proleg, but in the text the authors only mention caudal and cephalic to proleg; the authors also need to mention in the text that the proleg was also injected. We have modified this sentence to read, "We examined

whether injections with *P. aeruginosa* strain PAO1 slightly (approximately 1 mm) cephalic or caudal to the final proleg or at the site of the proleg altered LT_{50} values.”

27. *Additionally, because this is a methods paper, the authors should mention why the proleg is commonly as the injection site.* We have added the following sentence to the Supplemental Methods: “This site allows for deeper insertion of the needle without damaging critical internal structures and also facilitates dissemination of the inoculum (Serrano, et al. *Antibiotics*. 2023, 12,505. <https://doi.org/10.3390/antibiotics12030505>).”

28. *The authors should re-think the conclusion made in lines 226-227 that their findings highlight the "complex" relationship between..., and if the authors choose to keep it, they should expand on how exactly their findings highlight a complex relationship.* We have removed this sentence.

29. *The discussion repeatedly talks about how the present manuscript is important for P. aeruginosa research but fail to mention a broader application. Authors should not limit themselves to P. aeruginosa in the discussion and should mention the potential for its application in studies of other virulent bacteria beyond P. aeruginosa, particularly in the concluding paragraph of the discussion.* We have now revised the final paragraph of the Discussion to read: “In summary, we provide a protocol for accurately measuring and comparing the virulence of *P. aeruginosa* using *G. mellonella*. This approach may be useful in increasing our understanding of *P. aeruginosa* pathogenesis and the mechanisms by which this bacterium causes disease. It may also be applicable to other bacteria species, especially those that are highly virulent in *G. mellonella* (e.g., *Burkholderia pseudomallei*, *Burkholderia mallei* (32)) and for which the lethal dose methodology is not optimal. In addition, this methodology should be applicable to efficacy studies testing antibiotics and alternative therapeutics.”

30. *The authors say that smaller AUCs reflect more virulent strains- is this an assumption based on death of the larvae? Smaller AUCs with a plot of LT_{50} vs. $\ln(\text{dose})$ by definition mean that the larvae die faster at any given dose and require less time to achieve 100% mortality. It follows from this that smaller AUCs indicate higher virulence.*

31. *Is there known virulence of their clinical isolates samples that correlate to their results? Is it because of the mouse study? Please clarify.* The purposes of Figure 6 (the old Figure 7) and Table 1 are to address this very question. The figure and table show that there is a modest correlation between virulence measurements in *Galleria* and in mice.

32. *Lines 321-323: authors use "inbred" and "in-bred"- be consistent.* We have made this correction.

33. *Galleria are usually kept in a refrigerator to prevent metamorphosis. If this is the case for the present paper, authors should discuss how long they leave them out of the refrigerator before they assess movement for larval sorting in step 1 of supplemental methods.* We have added the following sentence to the Supplemental Methods: “Larvae may be stored in the dark at room temperature or cooler temperatures (4-15°C). Cooled larvae should be rewarmed to room temperature for 1-2 hr prior to use.”

34. *The supplemental methods section describes how to dilute sample to obtain CFU/mL but does not mention that the researcher has to plate the dilutions and count colonies. Please add.* This is described in the Supplemental Methods section entitled, “Enumeration of Bacterial Numbers in Inoculums.”

35. *In the supplemental methods, the authors should explain the reason for various steps. One example is "ensure that the bevel of the needle is facing upwards when..." - why? This is a standard injection technique to ensure that just the sharp tip of the needle first punctures the surface of the larva when the syringe is held at an angle. If the bevel is down, then the beveled surface of the entire needle will make contact with the surface of the larva at about the same time.*

36. *In the supplemental methods, the statement "16. Note that..." requires a reference. We have added the following reference, which discusses melanization and *Galleria* death: Menard et al., Front. Cell. Infect. Microbiol. 2021. doi: 10.3389/fcimb.2021.782733*

37. *Additionally, worms may appear immobile when shaking the dish, but still exhibit mouth movements observable under a dissection microscope. Therefore, the authors should specify that they designated larvae as dead when no movement was observed after shaking, to prevent confusion that no movement unequivocally means they are dead. We have revised this sentence to read as follows: "Larvae are adjudicated as dead when they fail to show readily apparent macroscopic movements following gentle shaking of the Petri dish."*

Reviewer 2

1. *Figure 1 may need to be larger, and it would be helpful to show the injection point here.* Based on Reviewer 1's suggestion, we have deleted Fig 1. A photo showing the injection point is now in the Supplemental Methods, Protocol Fig 4. A photo showing an injection is now shown in the Supplemental Methods, Protocol Fig 3.

2. *Do the researchers intend to aim precisely for either the left or right proleg? We routinely inject into the left proleg for the sake of consistency. However, we did compare injection into the left proleg and the right proleg, and did not observe a difference in larval mortality. For this reason, we did not specify right vs. left in our protocol.*

3. *Being unfamiliar with injecting insect larvae, is the proleg a soft structure that can be injected easily? Yes, this is correct. It is also adjacent to a portion of the larvae that does not contain critical structures that could be injured by injection.*

4. *The supplement figure S3 shows an arrow pointed slightly adjacent to the proleg. Is there a recommended angle to hold the larva? It is difficult to determine from the small photo and top-down view. The angle between the needle and the surface of the larva is usually between 30-45 degrees. We have added this to the Supplemental Methods.*

5. *The use of highly virulent clinical isolates is a reasonable choice for this study. Figure 4 uses lab strain PAO1. Is it feasible to include PAO1 in Figure 6? It would be interesting to directly compare if lab strains have similar LT50 values to clinical isolates. We agree this is a very interesting question. However, we had hoped to limit the scope of this manuscript to describing a protocol for doing time-to-death experiments in *Galleria*. Our plan was to publish a second manuscript in which we used this protocol to answer several interesting questions about *P. aeruginosa* virulence, such as how laboratory strains compare to clinical strains in virulence, whether certain sequence types of *P. aeruginosa* are highly virulent, and whether highly antibiotic-resistant strains of *P. aeruginosa* are more or less virulent than antibiotic-susceptible strains. We feel that the very interesting question about PAO1 virulence would be better addressed in this second manuscript. In addition, because of Reviewer 1's suggestion (point 7), we have removed the PAO1 experiment from this manuscript and instead for consistency have*

now performed the site of injection experiments using PABL089 so that all the initial experiments shown in Figures 1, 2, and 3 use the same *P. aeruginosa* strain.

6. On line 270, Usage of comparing antimicrobial compounds is mentioned. How are antibiotic compounds or other anti-virulence drugs administered to *G mellonella*, and how would this model be used to study these types of effects? *Galleria* are frequently used for these purposes (see Menard et al., 2021. Front. Cell. Infect. Microbiol. doi: 10.3389/fcimb.2021.782733 for a nice review.) Usually, the antibiotic or alternative agent (e.g., phage) is injected shortly after the bacteria are injected. The second injection is done in the last proleg on the other side of the larva from the bacterial injection. We have now added a sentence to the final paragraph of the Discussion to list this as another potential application of this methodology.

7. Line 290 and following, discusses factors that may contribute to varying slopes of the lines, and speculates these may differ between *P. aeruginosa* strains. Is there existing literature using *G. mellonella* studies that supports any of these possibilities? To our knowledge, we are the first group to compare dose-response curves for infected *Galleria*, so we cannot support these conjectures with data. However, we are currently comparing the growth of different *P. aeruginosa* strains within *Galleria* to address this question and hope to be able to provide some mechanistic insights into this very interesting question in a future publication.

8. Line 56, assume "wax" is intended. Yes! Thank you.

9. Figure 2A requires higher resolution. We have included higher resolution figures for the resubmission.

Re: Spectrum01666-24R1 (Refined methodology for quantifying *Pseudomonas aeruginosa* virulence using *Galleria mellonella*)

Dear Dr. Alan R. Hauser:

Your manuscript has been accepted, and I am forwarding it to the ASM production staff for publication. Your paper will first be checked to make sure all elements meet the technical requirements. ASM staff will contact you if anything needs to be revised before copyediting and production can begin. Otherwise, you will be notified when your proofs are ready to be viewed.

Sincerely,
Mariola Ferraro
Editor
Microbiology Spectrum

Reviewer #2 (Comments for the Author):

The authors have responded sufficiently to my comments. I have no additional concerns.